# The Bidirectional Relationship between Pulmonary Tuberculosis and Lung Cancer

**DOI:** 10.3390/ijerph20021282

**Published:** 2023-01-10

**Authors:** Mădălina Preda, Bogdan Cosmin Tănase, Daniela Luminița Zob, Adelina Silvana Gheorghe, Cristian Virgil Lungulescu, Elena Adriana Dumitrescu, Dana Lucia Stănculeanu, Loredana Sabina Cornelia Manolescu, Oana Popescu, Elmira Ibraim, Beatrice Mahler

**Affiliations:** 1Marius Nasta Institute of Pneumology, 050159 Bucharest, Romania; 2Microbiology, Parasitology and Virology Discipline, Department of Fundamental Sciences, Faculty of Midwives and Nursing, “Carol Davila” University of Medicine and Pharmacy, 020021 Bucharest, Romania; 3Department of Thoracic Surgery, Institute of Oncology “Prof. Dr. Al. Trestioreanu” Bucharest, 022328 Bucharest, Romania; 4Department of Medical Oncology II, Institute of Oncology “Prof. Dr. Al. Trestioreanu” Bucharest, 022328 Bucharest, Romania; 5Department of Oncology, “Carol Davila” University of Medicine and Pharmacy, 020021 Bucharest, Romania; 6Department of Medical Oncology I, Institute of Oncology “Prof. Dr. Al. Trestioreanu” Bucharest, 022328 Bucharest, Romania; 7Department of Oncology, University of Medicine and Pharmacy Craiova, 200349 Craiova, Romania; 8Department of Virology, Institute of Virology “Stefan S. Nicolau”, 030304 Bucharest, Romania; 9Pneumo-Phthisiology II Discipline, Faculty of Medicine, “Carol Davila” University of Medicine and Pharmacy, 020021 Bucharest, Romania

**Keywords:** lung cancer, tuberculosis, immune checkpoint inhibitors, carcinogenesis

## Abstract

Lung cancer and pulmonary tuberculosis are two significant public health problems that continue to take millions of lives each year. They may have similar symptoms and, in some cases, are diagnosed simultaneously or may have a causal relationship. In tuberculosis disease, the chronic inflammation, different produced molecules, genomic changes, and fibrosis are believed to be important factors that may promote carcinogenesis. As a reverse reaction, the development of carcinogenesis and the treatment may induce the reactivation of latent tuberculosis infection. Moreover, the recently used checkpoint inhibitors are a debatable subject since they help treat lung cancer but may lead to the reactivation of pulmonary tuberculosis and checkpoint-induced pneumonitis. Pulmonary rehabilitation is an effective intervention in post-tuberculosis patients and lung cancer patients and should be recommended to improve outcomes in these pathologies.

## 1. Introduction

Lung cancer, the primary cause of cancer-related deaths worldwide, taking 1.8 million lives in 2020, is now recognized as a public health concern [1,2]. In 2021, a total of 2,206,771 newly diagnosed lung cancer cases were reported worldwide [3].

Another significant global public health issue is tuberculosis (TB), caused by *Mycobacterium tuberculosis* [4]. *M. tuberculosis* is an intracellular organism that can spread efficiently through the respiratory system of the general population by causing life-threatening lung disease in immune hosts [1]. TB, a contagious illness caused by *M. tuberculosis*, continues to be a leading cause of morbidity and mortality in the world [5]. *M. tuberculosis* infected one-third of the population in 2021, producing disease in approximately 10% and killing 1.6 million people [1,5,6]. The COVID-19 pandemic has been said to have erased years of advancement in TB control efforts because of lower notification and treatment trends and increased fatality reports [5]. Between 2019 and 2020, the number of people newly diagnosed with TB decreased by 18%, from 7.1 million to 5.8 million, after sharp increases between 2017 and 2019 [5]. In 2021, there was a slight increase, reaching 6.4 million [5].

Initial identification by macrophages is a key phase that mobilizes the immune response in both TB and cancer [7]. In the lungs, *M. tuberculosis* is endocytosed and trapped in the phagosome of an alveolar macrophage [7]. T cells need to be stimulated intensively to activate macrophages and eliminate mycobacteria later [7]. To survive these processes, *M. tuberculosis* has developed unique mechanisms that either detoxify the toxic compounds before harm is done or repair the damage before it becomes fatal [7]. Additionally, it is known for *M. tuberculosis* to evade the phagosome and survive in the macrophage’s cytoplasm [7]. This results in granuloma, the TB disease’s defining feature [7].

Numerous studies have demonstrated how viruses cause genomic instability, which leads to cellular change [8]. This is accomplished by regulating a wide range of cellular genes and signaling pathways involved in immune responses, proliferation, and oncogenesis [8]. Other microorganisms besides viruses have a role in this pathogenesis [8].

Many theories, including the suppression of the immune system, DNA damage, and the production of inflammatory factors, suggest that *M. tuberculosis* infection can lead to lung cancer in some cases [1]. The essential premise is that *M. tuberculosis* causes chronic inflammation, which in turn may lead to lung cancer [1].

## 2. Tuberculosis as a Risk Factor for the Development of Lung Cancer

The two-stage carcinogenesis model can be used as a basis for the theory of involvement of chronic inflammatory processes in the tumorigenic process [9]. The beginning event, such as a genotoxic damage, is succeeded by promoting events and conditions that cause the clonal expansion of altered cells [9]. The propagation of a cell with genomic injury is caused by the accumulation of mutations and cell viability, which may be the basis for carcinogenesis [8]. Furthermore, certain intracellular bacteria have been shown to induce resistance to cell death to increase their own longevity [10]. For instance, the activation of RAF kinases and the WNT signaling pathway by interacting with ẞ-catenin leads to cell proliferation and transformation [10]. In contrast, the inhibition of BAX and overexpression of the Bcl2 family lead to the abrogation of apoptosis [10]. Both of these mechanisms, when left unchecked, can result in the hallmarks of cancer [10]. Due to their relationship with lung inflammation, pulmonary infections can theoretically contribute to the growth of lung cancer [11]. It is well-known that *M. tuberculosis* modifies these signaling pathways to aid its survival [12]. The severe chronic lung infection tuberculosis, which is brought on by *M. tuberculosis*, is possibly the best researched of them [11]. Although the proportional importance of these factors in promoting any particular cancer is unclear, mounting evidence points to chronic inflammation as a potential genotoxic environment and a major promoter of the carcinogenic process [9].

The resident alveolar macrophages are among the first cells to come into contact with the aerosolized *M. tuberculosis* in the lungs [13]. They engulf the bacilli in phagosomes that may combine with acidic lysosomes [14]. *M. tuberculosis* manipulates the phagolysosome maturation process to avoid lysosomal sequestration and death so that it can survive and reproduce inside phagosomes [14]. *M. tuberculosis* lives in the phagosome, where it multiplies; it breaches the phagosomal barrier and escapes into the macrophage’s cytoplasm, where it can multiply or spread through the necrosis of the infected cells, infecting other cells [13]. However, the specific mechanisms through which *M. tuberculosis* is able to replicate in the cytosol are mainly unknown and require other studies [13].

There is evidence to show that active innate immune responses in local macrophages are linked to early clearance of *M. tuberculosis* infection [15]. In addition to the lung-resident macrophage subsets, the recruited monocytes and monocyte-derived macrophages have been proposed to have a protective function during *M. tuberculosis* infection [13]. Due to its distinct cell surface lipids and secreted protein effectors, *M. tuberculosis* is able to avoid being killed by innate immune cells and selectively colonize alveolar macrophages [13]. Most alveolar macrophages in the lungs harbor *M. tuberculosis* after an aerosol-based infection [15]. To withstand immunological or pharmacological pressure, *M. tuberculosis* can acquire and maintain a slowed metabolic activity in the latent infection phase [15].

It has been suggested that TB-related chronic lung inflammation may induce clastogenic activity in the bronchial epithelium’s DNA [8]. As an intracellular organism, *M. tuberculosis*, bacterial DNA may integrate to bronchial epithelial cells, causing neoplastic transformation [8]. Lateral gene transfer is another possibility [10].

By causing E-cadherin expression to change into epithelioid morphology, *M. tuberculosis* can connect macrophages during the development of granulomas, a process similar to the mesenchymal–epithelial transition (MET) that takes place throughout malignancies [8]. Additionally, *M. tuberculosis* and other persistent infections activate transcription factors that trigger the MET, a crucial step in the development of cancer and metastatic disease [8].

Both TB and lung adenocarcinoma overexpressed MKI67, a nuclear protein that controls the regulation of several genes to promote tumorigenesis [16]. Moreover PtpA, a significant effector protein that can enter the nucleus of the host cell and control cell migration and proliferation to encourage tumor growth, seems to support MKI67-mediated tumor cell invasion, migration, and proliferation during *M. tuberculosis* infection [16].

While not necessary for *M. tuberculosis* growth in vitro, *M. tuberculosis* PtpA is a secreted, low-molecular-weight protein tyrosine phosphatase (PtpA) that is significant for *M. tuberculosis* pathogenicity in vivo [17]. In addition, PtpA may not be the only *M. tuberculosis* effector protein that targets MKI67 to control these activities [4]. Moreover, host genes may be implicated in the proliferation, migration, and invasion of tumor cells induced by *M. tuberculosis* [18]. In a mouse xenograft model and in vitro, PtpA-expressing *Mycobacterium bovis* BCG promotes the growth and migration of human lung adenoma A549 cells [18].

It is well-known that TB-related lung fibrosis and inflammation can cause genetic damage, which may raise the chance of developing lung cancer [19]. Additionally, extracellular matrix elements like collagens are essential for promoting lung cancer [17]. In a study including 35 patients with TB, 48 lung adenocarcinomas, and 21 with sarcoidosis, the most prevalent pathogenic trait that TB, lung cancer, and sarcoidosis had in common was lung matrix remodeling with fibrillar collagen deposition [4]. There were higher levels of type I and type III collagens in the lung lesions of patients with TB, lung adenocarcinoma, and sarcoidosis [4]. This is consistent with clinical observations that these diseases are typically accompanied by fibrotic progression and, as a result, impaired lung architecture and function [17].

A systematic review based on the risk of lung cancer after tuberculosis showed an increased risk when research was limited to people from “non-Westernized countries” [19]. There was a statistically significant 1.78-fold rise in the incidence of lung cancer among TB patients when only the never-smokers were selected [19]. A 1.74-fold increase is consistent with the pattern of findings seen across studies that account for confounding by cigarette smoking [19]. After correcting for lifetime exposure to environmental cigarette smoke, the intensity of this connection increased to 2.93-fold [19]. The increased risk of lung cancer was greatest in the first five years following a TB diagnosis, but the risk persisted at 1.99 times higher for more than 20 years [19]. Even after more than 20 years from the diagnosis of TB, there has been a considerable increase in the risk of lung cancer [19].

The risk of having TB was shown to be correlated with cancer in a time-dependent manner, peaking 1 year before and after cancer diagnosis [16]. In a study evaluating tuberculosis risk in lung cancer patients, although TB prevalence was significantly higher among men than women, the risk factors were similar [16].

## 3. Early Misdiagnosis of Cancer as Tuberculosis and Vice Versa

Common signs of pulmonary TB and lung cancer include a cough, expectoration, fever, hemoptysis, weight loss, and shortness of breath [20]. Due to the similarity of their symptoms, lung cancer and TB can be challenging to diagnose [20]. Knowing the characteristics of each pathology is important since patients with lung cancer could be given the wrong diagnosis of pulmonary TB, delaying the right diagnosis and subjecting them to the wrong treatment and vice versa [20]. The five primary manifestations of TB on a chest scan are cavitation, pleural effusion, miliary disease, lymphadenopathy, and parenchymal disease [20,21]. The most frequent radiographic finding in lung cancer is a mass with or without collapse [20]. The edges of malignant tumors are uneven and feature radiating threads. Hilar prominence, pulmonary nodules, mediastinum expansion, entire or partial atelectasis of a segment, lobe, or lung, unresolving consolidation, cavitation, raised diaphragm, or pleural effusion are further signs of lung cancer [20]. Rib erosion and lymphangitis are some other findings [20]. In 0.4% of lung cancer cases, a normal chest X-ray is discovered [20].

High CA 19-9 levels as a tumor marker for gastrointestinal, pancreatic, and hepatobiliary cancers are suggestive of advanced illness and a bad prognosis [22]. Overexpression of CA 19-9 has been seen in some benign conditions besides cancer, such as gastrointestinal problems, hepatobiliary system diseases, pneumonia, pleural effusion, renal failure, and systemic lupus erythematosus, and this phenomenon may be related to glycan-mediated cell–cell interactions in mucosal immunity [22]. This indicator’s specificity is therefore limited [23]. Sharply elevated CA 19-9 levels are also important in the misdiagnosis of TB and cancer [23]. In instances of pulmonary TB, hematogenously disseminated tuberculosis, and pancreatic tuberculosis, CA 19-9 was apparently increased [23]. Similarly, CA 125 and adenosine deaminase activity are not specific in differentiating cancer of tuberculosis [22]. The three markers CEA, CYFRA21-1, and NSE work better together to detect lung cancer patients who are likely to receive the incorrect diagnosis of pulmonary TB [24].

In a study, 6683 individuals had been diagnosed with TB in total [25]. Among them, 978 (14.6%) and 45 (0.7%) of the original diagnoses had been changed to lung cancer and non-TB reasons, respectively. This study discovered that early pathology acquisition might aid in earlier revision of diagnosis. Patients with lesions that are >3 cm in size, a radiologic miliary pattern, no anti-TB therapy, and cultures that are negative for NTM may prompt doctors to schedule a pathology study. Contrarily, those who have a lesion size of less than 3 cm, no radiologic miliary pattern, anti-TB treatment, and cultures positive for NTM are more likely to experience a delayed diagnosis. As a result, our findings highlight the importance of keeping lung cancer in mind in this subgroup and obtaining earlier pathology whenever possible.

## 4. Checkpoint Inhibitors in Tuberculosis and Cancer

Protective immunity against pathogens such as viruses, bacteria, fungi, parasites, and malignancies is mediated by different subsets of effector T cells, such as CD4^+^ Th1, Th2, and Th17 cells, as well as CD8^+^ cytotoxic T lymphocytes [26,27]. These effector T cells must be strictly controlled since they can cause both acute and chronic inflammation, which can result in immunopathology or autoimmune disease [26].

Chronic infections like tuberculosis are comparable to cancer in that they both involve prolonged periods of high antigen exposure [28,29]. The most important defense mechanism for destroying intracellular mycobacteria in the early stages of *M. tuberculosis* infection is immune response activation with type 1 T helper cells (Th1) and production of IFN-ɣ and TNF-α [26].

The expression of PD-1 on CD4 and CD8 T cells does rise after prolonged exposure to tuberculosis antigens, and inhibiting PD-1 or PD-L1 using an antibody boosts CD4 and CD8 T-cell activity [28]. It is currently believed that T-cell depletion or malfunction is a major factor in the decreased T-cell defenses against infections or malignancies [26]. From a molecular perspective, it makes sense to assume that anti-exhaustion medications will enhance the immune response to tuberculosis because T-cell exhaustion has been demonstrated to limit the functionality of ESAT-6-specific CD4 T cells during *M. tuberculosis* infection [30]. T cells with a similar profile were later found in the tumor microenvironment, though exhausted T cells were initially discovered in the context of persistent infections [26]. T-cell depletion causes immune responses to be inefficient in the tumor microenvironment. Immune checkpoint inhibitors with targets like programmed cell death protein 1 (PD-1), programmed death ligand 1 (PD-L1), and cytotoxic T-lymphocyte-associated protein 4 (CTLA-4) have been the mainstay of successful tumor immunotherapy treatments until recently [7].

The interaction between co-stimulatory and co-inhibitory receptors on T-cell surfaces (such as CD28 and CTLA-4), as well as CD80 (B7-1) and CD86 (B7-2) on antigen-presenting cells, is a crucial step in the T-cell-mediated immune response [31]. Regulatory T (Treg) cells and immunological checkpoints like CTLA-4 and PD-1 play a significant role in suppressing anti-tumor T-cell responses [31]. Additionally, it is thought that blocking immunological checkpoints and immune evasion play a role in allowing MTB to form latent infections [31].

Checkpoint inhibitors are a rapidly expanding field of study in cancer immunotherapy, with potential applications to tuberculosis [7]. Immune checkpoints are significant in that they have unique mechanisms and nonredundant functions, highlighting the intricacy of the regulation of T-cell responses [26]. Immune checkpoint inhibitors, mostly monoclonal antibodies against immune checkpoints on immune cells that disrupt the interaction with the corresponding checkpoint ligands, have been approved for use in humans with cancer, most notably melanoma [26]. An immune checkpoint blockade’s primary goal is to lessen the inhibition of effector T cells, particularly CD8^+^ T cells, and to enhance tumor-specific immune responses as a result [26]. The main idea is that by carefully suppressing immune regulatory pathways, an infection or tumor can be cleared by the immune system more effectively [7]. They restore anticancer immune responses by impairing co-inhibitory T-cell signaling, as seen in their effects on programmed death 1 (PD-1) and programmed death ligand 1 with pembrolizumab, nivolumab, and atezolizumab, respectively [32]. Additionally, an immune checkpoint blockade may increase the efficacy of other immunotherapeutic strategies, such as cancer vaccines, by removing the brake on T-cell suppression and so increasing the T-cell effector function [26]. Most of these inhibitory receptors have been the subject of antibody development, testing, and clinical trials; several of these have now received human-use authorization [26]. The ability of PD-1 inhibition to prevent effector T cells from dying during lung infection is also promising [33]. One of the main contributing factors to the alarming rise in TB is immunosuppression, sometimes brought on by the decrease in CD4^+^ T cells, as it occurs in HIV infection [34].

An essential factor of the interplay between hosts and pathogenic microorganisms is the inhibitory PD-L/PD-1 pathway [34]. In pulmonary TB patients compared with control subjects, a significantly increased expression of the levels of PD-1 was observed, as well as its two ligands PD-L1 and PDL-2, on CD3^+^ T cells [34]. This may indicate that *M. tuberculosis* antigen significantly activates PD-1 signaling pathways for both CD4 and CD8 T-cell immune response [31]. *M. tuberculosis* infection promotes the tumor metastases in lung tissue, PD-1 signaling being required in this regulation [31].

The PD-1/PD-L1 pathway’s function in MTB infections is debatable because of inconsistent results from mouse and human studies [26]. According to some theories, MTB suppresses immunological reactions by encouraging PD-L1 expression on DCs, which boosts the induction of Treg cells [26]. However, increased inflammatory response and unchecked bacterial proliferation in the lungs made PD-1^−/−^ animals more vulnerable to MTB-induced death [26].

An inhalation infection model was used to compare PD-1-deficient and control C57BL/6 mice to examine the function of PD-1 in aggressive TB [34]. At different periods after infection, the survival and bacterial growth in the lungs, liver, and spleen were observed [34]. Surprisingly, animals lacking PD-1 had diminished resistance to *M. tuberculosis* H37Rv aerosol infection [34].

In contrast to control mice, which exhibited fewer and less severe inflammatory lesions, *M. tuberculosis* H37Rv-infected animals with PD-1 deficiency had large localized necrotic regions in their lungs [34]. Both PD-1^−/−^ and wild-type mice developed extremely uncommon inflammatory nodules two weeks after infection [34]. However, all groups experienced consolidating nodular pneumonia 4 weeks after infection, but the type of inflammation was noticeably different [34]. The granulomatous multifocal pneumonia in wild-type mice was extensive but not severe, and the nodular aggregates were mostly histiocytic with infiltrating lymphocytes and little necrosis [34]. In contrast, PD-1^−/−^ mice developed necrotizing pneumonia with considerable lymphocytic infiltrates but few degenerating and fragmented neutrophils [34]. Extensive necrosis was linked to neutrophil infiltration mixed with dying macrophages [34]. Without severe degradation of the respiratory epithelium, nodules seemed to form around bronchioles, and many bronchioles were filled with or had dispersed neutrophils and nuclear and cellular debris [34]. The widespread lung parenchymal necrosis and architectural loss in the cores of these nodules were replaced by nuclear and cellular debris [34]. The reduction in IL-2 production, decreased proliferative potential, decreased cytotoxic capacity, and impaired generation of pro-inflammatory cytokines are all functional traits of exhausted T cells. Increased expression of several immunological checkpoints, including PD-1, CTLA-4, lymphocyte activation gene-3, T-cell immunoglobulin and ITIM domain, and T-cell immunoglobulin-3, is a key indicator of T-cell fatigue [26]. It has been shown that elevated immunological checkpoint expression causes a decline in T-cell function [26]. Immune checkpoints and their ligands interact in a complex way at various stages of T-cell activation and activity [26]. In the T-cell priming stage, for instance, CTLA-4, LAG-3, TIM-3, and TIGIT predominantly engage with their ligands, preventing T-cell activation [26]. However, PD-1 expression is increased on activated T cells, and ligation of PD-1 with PD-L1 or PD-L2 primarily occurs in the periphery, which results in the inhibition of activated T cells at effector phase 6 [26].

Evidence suggests that in the event of a persistent *M. tuberculosis* infection, blocking PD-1/PD-L1 is insufficient to restore effector T-cell activity [35]. This is thought to be caused by an immune system that lacks homeostasis, which causes immunological hyperactivation and inflammatory damage [35]. Furthermore, T cells that express PD-1 during *M. tuberculosis* infection are not exhausted T cells in the traditional sense, since they contribute to the maintenance of effector T cells that are specific for the antigen [32]. T-cell exhaustion-related epigenetic remodeling is likewise unaffected by PD-L1 inhibition and, if antigen levels remain high, can result in a return of the condition [7]. As a result, populations that are worn out are unable to trigger memory responses [7]. Some diseases do not respond to PD-1 blocking, such as head and neck squamous cell carcinoma, which has response rates as low as 10–15% [28]. As a result, alternate immunotherapies utilizing different checkpoint inhibitor agents are required to reverse T-cell depletion in conditions including TB and head and neck squamous cell carcinoma [28].

Another immunotherapy target is T-cell immunoglobulin and mucin domain-containing-3 (TIM3) [30]. Contrary to PD-L1, blocking TIM3 is expected to improve the outcomes of *M. tuberculosis* infection, and this effect is thought to be mostly mediated by the recovery of worn-out T cells [30]. Furthermore, it has been discovered that head and neck squamous cell carcinoma TIM3 expression is actually increased by PD-1 inhibition [30]. For both disorders, a combination of strategies involving a dual inhibition of PD-1 and TIM3 may be necessary to overcome T-cell depletion [30].

Without PD-1, the inflammatory and necrotic responses to *M. tuberculosis* are significantly amplified, demonstrating the crucial function of this co-inhibitory receptor in regulating inflammatory reactions to a highly immunogenic pathogen [34]. When the PD-1 pathway is rendered inoperable by gene deletion, *M. tuberculosis* infection is lethal, as opposed to the outcomes seen in chronic infection models, where blockage of PD-1 produced higher effector activities in T cells and a more robust response against the pathogen [34].

Immune checkpoints that regulate T-cell activation and regulatory cells of the innate and adaptive immune systems, such as Treg cells, myeloid-derived suppressor cells (MDSCs), and M2-type macrophages, all play a role in immune regulation [26]. As a mechanism of immune subversion, these regulatory cells, molecules, and immune checkpoints are frequently heightened during cancer and chronic infections. As a result, they have emerged as very important therapeutic targets in the treatment of cancer [26]. They also can potentially improve the efficacy of cancer and infectious-disease vaccines [26].

## 5. Managing the Coexisting Tuberculosis and Lung Cancer

There is a connection between the two processes—lung cancer and TB [17]. Lung cancer risk factors include epithelial dysplasia, bronchial and alveolar deformation, and post-tubercular alterations [17]. Old tuberculosis foci reactivate, and the *M. tuberculosis* spreads as lung cancer progresses [17].

At this moment, the specific guidelines recommend that all patients, including those with HIV co-infection, who have not received treatment before and who do not exhibit signs of drug resistance, should be given an internationally recognized first-line treatment regimen utilizing medications with known bioavailability [36]. Two months of isoniazid (H), rifampicin (R), pyrazinamide (Z), and ethambutol should represent the first phase (E) [36,37]. For the duration of the next four months (2HRZE/4HR), isoniazid and rifampicin should be administered [36,37]. It may be easier to administer medications when they are combined in fixed doses of two (isoniazid and rifampicin), three (isoniazid, rifampicin, and pyrazinamide), or four (isoniazid, rifampicin, pyrazinamide, and ethambutol) [36]. Ethambutol (or streptomycin, if it is used as a fourth medication) can be stopped once it is established that the TB isolate is fully sensitive [38]. Unfortunately, first-line anti-TB medication resistance is enhanced in *M. tuberculosis* multidrug-resistant (MDR) strains, and the expansion of resistance to second-line anti-TB therapies reduces the range of available treatments [39,40]. Knowing the risk factors for MDR-TB is crucial for the proper and timely start of treatment as this is a global health concern [39,40]. There are certain differences in the diagnosis of lung cancer in individuals with tuberculosis or who still have symptoms of the disease [17]. These depend on the various clinical symptoms brought on by the coexistence of diseases, the clinical course, and the location of the tumor (both processes are located in the same region or separately) [41]. When active tuberculosis coexists with cancer, common symptoms include patients’ general health getting worse (fever, dyspnea, often sputum with blood, weight loss, and others) [41]. Investigations have revealed that the X-ray image in these cases is varied: there are foci of infiltration of varying sizes, atelectasis, hypoventilation of different lung regions, and foci of destruction with cavern development [41].

In a study including 84,984 patients with lung cancer diagnosed over a period of 8 years, 8015 were diagnosed with tuberculosis before or at the time of cancer diagnosis [42]. For the 1155 patients diagnosed with tuberculosis after the lung malignancy diagnosis, the treatment completion was followed up [42]. Comparing the treatment groups, the clinical stage of lung cancer was higher in the incompletion group and stage IV was more common (56.09% vs. 36.97%) [42]. Lung surgery was more common in the group that finished their therapy than in the group that did not (39.42% vs. 22.66%) [42]. Rifampicin, isoniazid, ethambutol, and pyrazinamide, first-line anti-TB drugs, were administered more often in the treatment completion group than in the treatment incompletion group [42]. The death rate in the therapy non-completion group was greater at 1 year (67.56% vs. 20.49%) and overall since TB diagnosis (87.82% vs. 66.82%) [42]. A total of 61.13% of lung cancer patients who were also confirmed with new-onset TB did not finish their TB therapy, which is significantly higher than the typical incidence of unfinished TB treatment [42].

Medical treatment options for lung cancer include chemotherapy, radiation, immunotherapy, and targeted therapy [42]. These treatments might lead to drug interactions and worsen the adverse effects of anti-TB drugs, including skin rashes, hepatitis, and renal impairment [42]. Additionally, cancer-related frailty may prevent TB therapy from being finished, leading to problems like recurrence and increased mortality [42].

Patients with lung cancer, especially those with non-small-cell lung cancer, have greatly benefited over the past 10 years from immunotherapies that manipulate the immune system to find and eliminate cancerous cells [43]. In addition to chemotherapy, targeted therapy, radiation, and surgery, antibodies targeting inhibitory signaling receptors expressed on immune and tumor cells are now one of the key treatment options for patients with lung cancer [43]. The immune checkpoint inhibitors approved by the European Medicines Agency (EMA) in lung cancer, together with their recommendations, are summarized in Table 1.

Pulmonologists actively monitor and work with patients with lung cancer from the time of their first diagnosis and staging through treatment and restaging [43]. In the beginning, a multidisciplinary team will frequently include the thoracic oncology surgeon [43]. Pulmonologists are involved in the management of comorbidities and, more crucially, in issues caused by the cancer itself or the treatment, in addition to palliative and end-of-life care [43]. They ought to work with the oncologist to treat any lung-related morbidity that may have existed or developed as the patient had their oncological journey (including respiratory failure in the terminal stages) [43]. The interdisciplinary collaboration between pulmonologists and oncologists is crucial for therapy decisions because they are the authorities on lung physiology [43]. This is especially important because the most common respiratory comorbidity in patients with lung cancer, chronic obstructive pulmonary disease (COPD), affects their Eastern Cooperative Oncology Group (ECOG) performance status scale and must be considered when choosing the oncological treatment [43].

Even while immune checkpoint inhibitors have proven to be a highly effective treatment for advanced NSCLC, they only help a very small proportion of patients [49]. An immune checkpoint blockade is resistant to both primary and adaptive mechanisms, as evidence shows more and more frequently [49]. Additionally, a sizeable portion of patients who experience immune-related adverse events (irAEs) may require stopping [49].

Immune checkpoint inhibitors have the potential to cause severe pulmonary toxicity, which requires early diagnosis and treatment [49]. The pneumonitis caused by immune checkpoint inhibitors, also known as checkpoint-induced pneumonitis and immune checkpoint inhibitors-related interstitial lung disease, shares many overlapping lung symptoms with pneumonitis [49]. Furthermore, lung cancer patients appear to experience this severe, potentially deadly irAE more frequently [50]. Another critical collaboration between oncology and pulmonology, which will become more complex with the growth of immunotherapy, is to understand the immune-related adverse effects of these novel therapeutic approaches and their anticancer immunotherapeutic processes [50].

There are still many knowledge gaps, even though several standards have made it simpler to classify pneumonitis severity and standardize treatment approaches [51]. Potentially catastrophic irAEs, which are frequently challenging to treat, can cause a wide range of lung involvement, including acute symptoms, fibrotic lesions, and hypersensitivity reactions [51]. Due to the underlying lung cancer, other respiratory chronic disorders (COPD, asthma, and TB), or acute symptoms, the diagnosis of pulmonary irAE might also be difficult (pulmonary embolism, acute respiratory distress syndrome, and SARS-CoV-2 infection) [51].

Therefore, there is a pressing need to improve communication between oncologists and pulmonologists in the multidisciplinary care of lung cancer patients, particularly for those receiving immunotherapy with immune checkpoint inhibitors who run the risk of developing immune-related pneumonitis [51].

## 6. Pulmonary Rehabilitation for Lung Cancer and Tuberculosis

The therapy of patients with chronic respiratory disorders, such as COPD, bronchiectasis, and interstitial lung disease, among others, has been acknowledged to include pulmonary rehabilitation as a key component [52]. Recent studies indicate that lung cancer management may be positively impacted by pulmonary rehabilitation by enhancing several clinically significant outcomes, including performance status, fatigue caused by chemotherapy, oxygen consumption, exercise tolerance, and health-related quality of life [52].

An increasing amount of data also indicates the advantages of PR for patients with chronic lung impairment brought on by previously treated tuberculosis [53].

Patients report weight loss, asthenia, and exercise dyspnea during the active phase of TB because of the severe chronic inflammatory state that causes the release of many cytokines, TNF-α, and interleukins [54]. As a result of their high level of physical and mental stress, patients with TB should be included in PR programs (physical exercises, nutrition counseling, and psychological and emotional support) [54]. These patients’ complex medical recovery aims to achieve the highest level of independence and community reintegration following hospital discharge [54].

Quitting smoking at or around lung cancer diagnosis is also part of the PR programs [55]. It was significantly associated with improved overall survival, consistently among patients with lung cancer, irrespective of the histological type [55,56].

## 7. Conclusions

Pulmonary tuberculosis and lung cancer are at the top of the leading causes of morbidity and mortality worldwide, a situation worsened by the SARS-CoV-2 pandemic. Genomic modification, chronic inflammation, and fibrosis produced by tuberculosis may promote carcinogenic effects. Treatment used in lung cancer, for example, checkpoint inhibitors, may reactivate latent tuberculosis or aggravate its evolution. Special attention must be given to the treatment of coexisting pulmonary tuberculosis and lung cancer. Moreover, in both diseases, pulmonary rehabilitation plays a significant role, which in multiple situations is overlooked.

## Figures and Tables

**Table 1 ijerph-20-01282-t001:** The immune checkpoint inhibitors approved by the European Medicines Agency (EMA) in lung cancer.

Malignancy	Immune Checkpoint Inhibitor	Target	Indication
Non-small-cell lung cancer (NSCLC)	Pembrolizumab	PD-1	As monotherapy is indicated for the first-line treatment of metastatic non-small-cell lung carcinoma in adults whose tumors express PD-L1 with a ≥50% tumor proportion score (TPS) with no EGFR or ALK positive tumor mutations [44]. In combination with pemetrexed and platinum chemotherapy, is indicated for the first-line treatment of metastatic non-squamous non-small-cell lung carcinoma in adults whose tumors have no EGFR- or ALK-positive mutations [44]. In combination with carboplatin and either paclitaxel or nab-paclitaxel, is indicated for the first-line treatment of metastatic squamous non-small-cell lung carcinoma in adults [44]. As monotherapy is indicated for the treatment of locally advanced or metastatic non-small-cell lung carcinoma in adults whose tumors express PD-L1 with a ≥1% TPS and who have received at least one prior chemotherapy regimen [44].
	Atezolizumab	PD-L1	As monotherapy is indicated as adjuvant treatment following complete resection and platinum-based chemotherapy for adult patients with NSCLC with a high risk of recurrence whose tumors have PD-L1 expression on ≥50% of tumor cells (TC) and who do not have EGFR mutant or ALK-positive NSCLC [45]. In combination with bevacizumab, paclitaxel, and carboplatin, is indicated for the first-line treatment of adult patients with metastatic non-squamous NSCLC [45]. In combination with nab-paclitaxel and carboplatin, is indicated for the first-line treatment of adult patients with metastatic non-squamous NSCLC who do not have EGFR mutant or ALK-positive NSCLC [45]. As monotherapy is indicated for the first-line treatment of adult patients with metastatic NSCLC whose tumors have a PD-L1 expression ≥ 50% TC or ≥10% tumor-infiltrating immune cells (IC) and who do not have EGFR mutant or ALK-positive NSCLC [45]. As monotherapy is indicated for the treatment of adult patients with locally advanced or metastatic NSCLC after prior chemotherapy [45].
	Nivolumab	PD-1	In combination with ipilimumab and two cycles of platinum-based chemotherapy is indicated for the first-line treatment of metastatic non-small-cell lung cancer in adults whose tumors have no sensitizing EGFR mutation or ALK translocation [46]. As monotherapy is indicated for the treatment of locally advanced or metastatic non-small-cell lung cancer after prior chemotherapy in adults [46].
	Durvalumab	PD-L1	As monotherapy is indicated for the treatment of locally advanced, unresectable non-small-cell lung cancer (NSCLC) in adults whose tumors express PD-L1 on ≥1% of tumor cells and whose disease has not progressed following platinum-based chemoradiation therapy [47].
	Cemiplimab-rwlc	PD-1	As monotherapy is indicated for the first-line treatment of adult patients with non-small-cell lung cancer (NSCLC) expressing PD-L1 (in ≥50% tumor cells), with no EGFR, ALK, or ROS1 aberrations, who have locally advanced NSCLC and are not candidates for definitive chemoradiation, or metastatic NSCLC [48].
	Ipilimumab	CTLA-4	In combination with nivolumab and two cycles of platinum-based chemotherapy is indicated for the first-line treatment of metastatic non-small-cell lung cancer in adults whose tumors have no sensitizing EGFR mutation or ALK translocation [46].
Small-cell lung cancer (SCLC)	Atezolizumab	PD-L1	In combination with carboplatin and etoposide, is indicated for the first-line treatment of adult patients with extensive-stage small-cell lung cancer (ES-SCLC) [45].
	Durvalumab	PD-L1	In combination with etoposide and either carboplatin or cisplatin is indicated for the first-line treatment of adults with extensive-stage small-cell lung cancer (ES-SCLC) [47].

## Data Availability

Not applicable.

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
