# Peer review of "The Bidirectional Relationship between Pulmonary Tuberculosis and Lung Cancer"

_ijerph, 2023, doi:10.3390/ijerph20021282_

Round 1
Reviewer 1 Report
Preda M et. al provides a review of the relationship between pulmonary tuberculosis and lung cancer. The authors discuss how TB causes chronic inflammation, produces different molecules, genomic changes, and fibrosis that promotes carcinogenesis. On the other hand treatment of lung cancers with checkpoint inhibitors lead to the reactivation of pulmonary tuberculosis.
This review covers an important topic and addresses recent knowledge in the field.
The manuscript is well written.
Optional suggestion (This is a critique and not a concern): it would be interesting if the authors discussed in brief about early misdiagnosis of Lung cancer as tuberculosis (TB) and vice versa.
Author Response
Reviewer 1
Dear Reviewer,
Thank you very much for your recommendations and specific advices to improve our manuscript. We have tried to implement your recommendations. We have also made the changes recommended by the other reviewer. We have written below a point-by-point reply.
Preda M et. al provides a review of the relationship between pulmonary tuberculosis and lung cancer. The authors discuss how TB causes chronic inflammation, produces different molecules, genomic changes, and fibrosis that promotes carcinogenesis. On the other hand treatment of lung cancers with checkpoint inhibitors lead to the reactivation of pulmonary tuberculosis.
This review covers an important topic and addresses recent knowledge in the field.
The manuscript is well written.
Thank you for your appreciation.
Optional suggestion (This is a critique and not a concern): it would be interesting if the authors discussed in brief about early misdiagnosis of Lung cancer as tuberculosis (TB) and vice versa.
Thank you for your suggestion. We have added a short section on the difficulties of diagnosis and misdiagnosis of lung cancer as tuberculosis and vice versa, where we have discussed the clinical, lab and imagistic aspects which could lead to misdiagnosis or could help in establishing the correct diagnosis.
Reviewer 2 Report
This review “The bidirectional relationship between pulmonary tuberculosis and lung cancer” by Mădălina Preda has been written precisely and give adequate information in the context of pulmonary tuberculosis and lung cancer.
The only suggestion I have is to introduce a table in which they can show all the information together like Checkpoint inhibitors in tuberculosis and cancer.
Author Response
Reviewer 2
Dear Reviewer,
Thank you very much for your recommendations and specific advices to improve our manuscript. We have tried to implement your recommendations. We have also made the changes recommended by the other reviewer. We have written below a point-by-point reply.
This review “The bidirectional relationship between pulmonary tuberculosis and lung cancer” by Mădălina Preda has been written precisely and give adequate information in the context of pulmonary tuberculosis and lung cancer.
Thank you very much for your consideration.
The only suggestion I have is to introduce a table in which they can show all the information together like Checkpoint inhibitors in tuberculosis and cancer.
Thank you for your recommendation. We have introduced a table with the immune checkpoint inhibitors that are EMA-approved in lung cancer and also a paragraph with the anti-TB drugs used for treating tuberculosis.
Reviewer 3 Report
This is a very good study which is well supported with available literature
Some questions are there which need to be answered:
Q.1 Structure of some compounds which are currently used against lung cancer and tuberculosis should be included in the manuscript?
Q2: check reference format throughout the whole manuscript?
Q4: Some grammatical errors were also found throughout the manuscript.

Author Response
Reviewer 3
Dear Reviewer,
Thank you very much for your recommendations and specific advices to improve our manuscript. We have tried to implement your recommendations. We have also made the changes recommended by the other reviewer. We have written below a point-by-point reply.
This is a very good study which is well supported with available literature
Thank you for your kind words.
Some questions are there which need to be answered:
Q.1 Structure of some compounds which are currently used against lung cancer and tuberculosis should be included in the manuscript?
Thank you for your recommendation. In the section on managing tuberculosis and lung cancer we have added the compounds most often used in the treatment of lung cancer and tuberculosis, as recommended by the guidelines at this moment. We also added a table with the EMA-approved immune checkpoint inhibitors used in lung cancer, together with the target receptor and treatment recommendation.
Q2: check reference format throughout the whole manuscript?
Thank you very much for pointing this out. We have re-introduced all the references using Zotero references manager and we have chosen the MDPI format to ensure we have a uniform and correct layout of the references in the text and at the end.
Q4: Some grammatical errors were also found throughout the manuscript.
Thank you for highlighting this. We have proofread the manuscript and corrected some grammatical/misspelling errors.